# Soil Contamination by Heavy Metals and Radionuclides and Related Bioremediation Techniques: A Review

Yelizaveta Chernysh [1,2,]*[ ], Viktoriia Chubur [1,2], Iryna Ablieieva [1,3][ ], Polina Skvortsova [1], Olena Yakhnenko [1], Maksym Skydanenko [1][ ], Leonid Plyatsuk [1] and Hynek Roubík [2,]*[ ]

[1] Departament of Ecology and Environmental Protection Technologies, Sumy State University, Kharkivska st. 116, 40007 Sumy, Ukraine; chuburv@ftz.czu.cz (V.C.); i.ableyeva@ecolog.sumdu.edu.ua (I.A.); p.skvortsova@aspd.sumdu.edu.ua (P.S.); o.jakhnenko@ecolog.sumdu.edu.ua (O.Y.); m.skydanenko@pohnp.sumdu.edu.ua (M.S.); l.plyacuk@ecolog.sumdu.edu.ua (L.P.)

[2] Department of Sustainable Technologies, Faculty of Tropical AgriSciences, Czech University of Life Sciences Prague, Kamýcká 129, 16500 Prague, Czech Republic

[3] Department of Thematic Studies, Faculty of Arts and Sciences, Linköping University, SE-581 83 Linköping, Sweden

[*] Correspondence: e.chernish@ssu.edu.ua (Y.C.); roubik@ftz.czu.cz (H.R.)

**Abstract:** The migration of heavy metals and radionuclides is interrelated, and this study focusses on the interaction and complex influence of various toxicants. The rehabilitation of radioactively contaminated territories has a complex character and is based on scientifically supported measures to restore industrial, economic, and sociopsychological relations. We aim for the achievement of pre-emergency levels of hygienic norms of radioactive contamination of output products. This, in its sum, allows for further economic activity in these territories without restrictions on the basis of natural actions of autoremediation. Biosorption technologies based on bacterial biomass remain a promising direction for the remediation of soils contaminated with radionuclides and heavy metals that help immobilise and consolidate contaminants. A comprehensive understanding of the biosorption capacity of various preparations allows for the selection of more effective techniques for the elimination of contaminants, as well as the overcoming of differences between laboratory results and industrial use. Observation and monitoring make it possible to evaluate the migration process of heavy metals and radionuclides and identify regions with a disturbed balance of harmful substances. The promising direction of the soil application of phosphogypsum, a by-product of the chemical industry, in bioremediation processes is considered.

**Keywords:** toxicants; remediation; biosorption technologies; phosphogypsum





## 1. Introduction

The pollution of soils by toxicants of different natures and origins is a current issue, as it disrupts the homeostasis of ecosystems. The soil is the starting point of the food chain, where all nutrients accumulate. Therefore, one of the most dangerous types of pollution associated with radioactive contamination and heavy metal contamination requires significant efforts in soil remediation [1,2].

An analysis of radioactive contamination of the territory of Europe with cesium-137 shows that about 35% of radionuclide fallout after the Chernobyl radiation accident on the European continent is located in the territory of Belarus. The contamination of Belarusian territory with cesium-137 with a density greater than 37 kBq/m$^2$ amounted to 23% of the entire country's area; for Ukraine, ~5%. In Ukraine, more than 3.5 million hectares of forest land is radioactively contaminated by accidental emissions from the Chornobyl nuclear power plant. A complex set of factors determines the current radiation situation in radioactively contaminated forests, in particular, the density of radioactive soil contamination, the

composition of radionuclides, the physical and agrochemical properties of soils, etc., which determine the intensity of the biological circulation of radionuclides in ecosystems [3].

In research by Morooka et al. [4], areas affected by nuclear power plant (NPP) disasters are presented. Thus, 31 radioactive particles from surface soils were detected in an area 3.9 km northwest of the Fukushima-1 NPP. $^{134+137}$Cs had the highest activity ever recorded for Fukushima-1 NPP ($6.1 \times 10^5$ and $2.5 \times 10^6$ Bq per particle after decay correction until March 2011). Taking into account their large size (120 μm), the impact of these particles on human health will be minimal, including radiation during static skin contact [4].

The polluting of soils with heavy metals and radionuclides can be natural or anthropogenic. Furthermore, different heavy metals have a specific accumulation rate and bioavailability based on the physical and chemical properties of soils; therefore, they have a different biomagnification rate, impact on human health, and ecological risk level [5]. In this regard, it is important to identify the main sources, fate, and specific features in the distribution of heavy metals and radionuclides in soils.

A complex interplay of biogeochemical processes, affected by factors such as pH, clay content, and redox potential, controls the transport and chemical stability of metallic contaminants in soil and sediment deposits. The transfer of heavy metals from the soil to plants depends on quantity factors, intensity factors, and reaction kinetics. These factors represent indicators of the overall quantity of potentially available elements, the activity and ionic ratios of elements in the soil solution, and the rate of transition of elements from the solid phase to the liquid phase and within the roots of the plant. Physical clay (particles < 0.01 mm) and silt particles (particles < 0.001 mm), which have a higher absorption capacity compared to larger fractions, have the greatest impact on the radionuclide mobility in soils. The addition of a silt fraction from chernozem or sod–podzolic soils to sand reduces the accumulation of Sr in oats and wheat by 1.5–2 times, and this effect is more significant for $^{137}$Cs. The transfer of $^{90}$Sr from soil to plants is four times higher on sandy soils compared to loamy soils. Similarly, the transfer rates for $^{137}$Cs and $^{60}$Co are 100 times and 40 times higher, respectively, on sandy soils [6]. According to the sorption efficiency of these isotopes, the soil is arranged in the following order: sod–podzolic soils (Albeluvisols), grey soils (Calcisols), yellow soils, red soils (Ferralsols, Alisols, and Acrisols), chestnut soils (Kastanozems), and black soils (Chernozem). Substantial transfer of radiocaesium to plants in sandy and sandy loam soils with a low content of clay minerals and organic matter has been reported [7]. However, within the same soil group, the nature of the uptake of $^{137}$Cs into plants may vary depending on the absorption capacity of the soil, the content of macro and microelements, and the pH of the soil solution. The sorption of $^{137}$Cs in the soil depends on the clay mineral content of the soil and K-saturation.

This effect of fine soil fractions is associated with a stronger fixation of radionuclides in them, which, in turn, is due to a larger specific surface of clay and silt particles and changes in the chemical properties of the soil: the content of exchangeable cations and organic matter, as well as the absorption capacity, increases [6]. In general, the effect of soil properties on the biological rate of radionuclides can be described as follows: the transfer of radionuclides to plants increases with a decrease in the content of clay, silt, organic matter, and the absorption capacity in the soil [8,9].

Adsorbed radionuclides are more strongly retained by organic mineral complexes than when sorbed in minerals of a different nature [10,11].

Summarising several studies, two soil management directions can be outlined:

1.  Incorporating soil amendments can effectively fixate toxicants [12,13].
2.  Supplying the soil with deficient nutrients is a method that helps plants resist heavy metal stress [14,15].

The type of soil should be taken into account for its effective treatment. For example, on loamy soils, the use of almost all types of fertilisers will increase yields and reduce the level of radioactive substances in plant products. On poorly mineralised and hydromorphic soils, the absorption of some radioactive substances can sometimes increase with the application of mineral fertilisers. Research on new fertiliser compositions (also biosolids)

based on a combination of organic and mineral components of sustainable raw materials to increase the stability of soil–plant systems remains relevant [16,17]. Furthermore, resistance of the soil–plant system to radionuclides and heavy metals refers to the ability of the system to limit the mobility of chemical pollutants due to the inherent buffering properties of the soil, thus controlling the transition of the latter to the aerial part of the plant [18].

The migration of heavy metals and radionuclides is interrelated, and this study focusses on the interaction and complex influence of various toxicants. Therefore, this research aimed to review the problems of the rehabilitation of contaminated ecosystems and the areas of application of bioremediation processes for this purpose. According to the goal, the task was set as follows:

1.  Review of the state of ecosystems contaminated with heavy metals and radionuclides.
2.  Identification of the advantages and disadvantages of using biosorption technologies for the joint fixation of heavy metals and radionuclides.
3.  Substantiation of the possibility of using phosphogypsum for soil bioremediation.

## 2. Methodological Approach

To implement the objectives of the review, taking into account the analysis of the general scheme of the pollution cycle to structure the impact and means of reducing it, a bibliometric analysis was used using data from the Scopus and Web of Science databases. For the systematisation of data and their management, Mendeley software (Elsevier, Amsterdam, The Netherlands) was used.

Therefore, the methodological approach to the literature analysis consists of the following steps described in the flowchart in Figure 1.

| Bibliometric analysis | Systematization | Formalization |
|---|---|---|
| • Identification of the main trends in the field of research on remediation of soils contaminated with HM and radionuclides | • Summary of existing approaches for remediation involving bioprocesses of heavy metals and radionuclides fixation | • Theoretical substantiation of the advantages and disadvantages of the biosorption approach to remediation of soil ecosystems.<br>• Formation of an integrated approach on the synergistic basis of the effects of heavy metals and radionuclides. |

**Figure 1.** The methodological approach to the implementation of the topic review rehabilitation of contaminated ecosystems and the areas of application of bioremediation processes.

To validate the approach of using phosphogypsum in bioremediation, a comparative analysis of the elemental composition of phosphogypsum of various origins and locations was conducted in different regions of the world. This analysis was based on the results of research on Ukrainian phosphogypsum, as well as previous studies by other authors in different countries and regions around the world. This allowed for the synthesis of existing information on the subject and provided a rationale for recommending the genesis of suitable phosphogypsum for use in bioremediation processes. The main stages of the analysis are illustrated in Figure 2.

**Figure 2.** The methodological approach for the comparative review of phosphogypsum.

The ICP-OES method was used to analyse the elemental composition of phosphogypsum. The measurement protocol is shown in Table 1.

**Table 1.** Protocol to analyse the elemental composition of phosphogypsum.

| Step | Description |
| --- | --- |
| 1 Drying | Over-drying at 40 °C for 24 h |
| 2 Milling | Fraction size smaller than 1 mm |
| 3 Digestion | Ethos 1 (MLS GmbH, Leutkirch im Allgäu, Germany) microwave-assisted wet digestion system for 35 min at 210 °C |
| 4 Measurement | Inductively coupled plasma-atomic emission spectrometry (ICP-OES, Agilent 720, Agilent Technologies Inc., Santa Clara, CA, USA) |

## 3. Review of the State of Ecosystems Contaminated with Heavy Metals and Radionuclides

### 3.1. Sources of Radionuclides and Heavy Metals in the Ecosystem

Soil is a complex mixture and a non-renewable natural resource, as it can only be restored on a geological timescale. Heavy metals, unlike biological compounds, are rarely biodegradable and therefore accumulate in the environment. Heavy metals in the soil have a toxicological effect on soil microorganisms, leading to a decrease in abundance and activity [8,19]. The relatively long half-life of radionuclides contributes to their long-term presence in the environment, leading to various health complications, such as cancer [20].

Table 2 shows that the mentioned metals have a common anthropogenic source. These activities have led to increased concentrations of heavy metals in the soil, contributing significantly to their occurrence in the environment [21,22].

Table 2. Main sources of some heavy metals in soils.

| HM | Sources | Effects on Soil | References |
|---|---|---|---|
| Cd | Non-ferrous metal extraction, production of phosphate fertilisers, burning of fossil fuels, waste incineration, tannery industry, electroplating, and battery disposal. | The disruption of metabolic functions hinders enzyme activities, reducing the availability of N and S in the soil for crops. | [20,21,23–25] |
| Pb | Emissions from power generation, metallurgy, mechanical engineering, metalworking, electrical engineering, chemistry and petrochemistry, woodworking and pulp and paper industries, food industry, and construction-material production, as well as automotive transport. | Organisms' metabolic abnormalities affect soil enzymes and interrupt nutrient balance, reducing soil productivity. | [19,21,23,26–28] |
| Zn | Emissions from non-ferrous metallurgy, waste incineration plants, coal combustion, and tyre wear. | Phytotoxic effects on soil fertility, diminishing microbial biomass N; and lacking essential soil macronutrients, such as phosphorus. | [9,21,26,29,30] |
| Cu | Emissions of non-ferrous metallurgy enterprises; combustion of leaded gasoline, municipal incinerators, and copper mining residue. | Limited amounts of soil N and S hinder crop production. Inhibit β-glycosidase more than cellulose. Diminish microbial biomass N. | [21,26,27,31,32] |
| Hg | Emissions from non-ferrous metallurgy, fossil fuel burning, steel production, metal smelting. | Disruption of metabolic function in organisms. | [21,26,33,34] |
| As | Burning of fuel, emissions from power generation, production of construction materials, pharmaceutical and textile industry. As used in herbicides, insecticides, and desiccants. | Disruption of metabolic function in organisms. | [21,22,26,27] |
| Cr | Emissions from ferrous and non-ferrous metallurgy (alloying additives, alloys, and refractories) and mechanical engineering (electroplating). | Disruption of metabolic function in organisms. | [21,26,35,36] |
| Ni | Emissions from non-ferrous metallurgy, burning of fuel, waste incineration, and chemical industries. | Disruption of metabolic function in organisms. | [21,26,37–39] |

The application of mineral fertilisers contributes to the increase in these elements (Cd, Pb, etc.) in the soil. Cu, Cr, As, Hg, Mn, Pb, or Zn enter the soil, along with other toxic chemicals, such as pesticides. The application of a wide variety of biosolids, such as livestock manure, composts, and sewage sludge, to the soil unintentionally leads to the accumulation of heavy metals such as As, Cd, Cr, Cu, Pb, Hg, Ni, Se, Mo, Zn, Tl, Sb, etc., in the soil [40,41]. The extensive mining and smelting of Pb and Zn ore have resulted in soil contamination that poses a risk to human and ecological health [42].

*3.2. Monitoring of Radionuclides and Heavy Metals in Ecosystems and Impact on Humans: Ukraine CASE Study*

Monitoring radioactive substances and heavy metals in the environment is essential since pollutants can accumulate and migrate in the elements of the trophic chain. Soil is an indicator of the ecological state of the environment. Proper organisation of background monitoring of contaminated areas allows for an effective assessment of the state of environmental objects, development of methods for biological soil remediation, and prediction of the future state of the biological environment. Therefore, the authors investigated [43] the migration and accumulation of heavy metals and radionuclides in the most significant protected areas of the Transcarpathian region and identified the main possible factors that affect the environmental monitoring process. As stated in Savchuk et al. [44], years after the Chernobyl disaster, the environmental situation in the Polesie zone of Ukraine remains difficult, as confirmed by the increased content of heavy metals in feed, milk, beef, and

pork. The highest concentration of Pb was detected in coarse feed and sunflower cake and meal (2462 mg/kg and 1639 mg/kg); 41.9% and 60.0% of samples of these types of feed, respectively, had exceeded the maximum allowed concentrations of Cd.

The study revealed that coal mining in Jiangxi Province, China, causes radioactive uranium contamination and heavy metal contamination with zinc and cadmium in the soil, and the proposed in situ leaching method can be used to remediate contaminated soils, but with attention paid to the potential environmental risks to the soil [45]. The study by Mohuba et al. [46], conducted in the Thyspunt area of South Africa's Eastern Cape province, a potential site for a nuclear power plant, revealed elevated levels of radionuclides, including 238U, 235U, 234U, 226Ra, 232Th, and 210Pb, mainly in rock formations of shale and quartzite due to the natural geochemistry of these rocks. This indicates the potential health risks associated with the ingestion of groundwater commonly used in the area. The study by Baghdady et al. [47] in the Bahariya Oasis of Egypt, located near large iron mines, identified elevated levels of Ba, Cr, Cu, Fe, and V in cultivated soils and Al, Cr, Cu, and V in uncultivated soils, exceeding acceptable limits, with the highest concentrations recorded in the northern oases near iron mines, while the highest values of activity concentrations, i.e., 40 K, were recorded in uncultivated soils rich in evaporites. The study by Mitrovic et al. [48] observed a significant decrease in soil 137Cs activity levels over a ten-year study period (2007–2017) in Palilula, Belgrade, with values declining from 16 Bq/kg to 3.9 Bq/kg; and in Surcin, Belgrade, from 18 Bq/kg to 12 Bq/kg. The study also identified variations in soil heavy metal concentrations and attributed the primary source of radionuclides and heavy metals to the widespread use of mineral phosphate fertilisers in agricultural fields.

In the context of the analysis performed, it is possible to define the main aspects of the effectiveness of soil-monitoring implementation [49]:

- Availability of sufficient areas that are subject to minimal anthropogenic impact (for example, biosphere reserves, nature reserves, and national nature parks);
- Selection of background monitoring criteria that would take into account the prevalence of individual substances in nature, their migration in the natural environment, and the presence of potential sources of their anthropogenic intake;
- Selection of effective methods for monitoring the state parameters of environmental objects.

The impact of the Chornobyl accident is not limited to the exclusion zone. Studies were carried out in different regions of Ukraine and protected areas to establish the migration processes of radionuclides and heavy metals and the possible relationships between them. The determination of the heavy metal content in soils in the Carpathian Mountains region and bottom sediments and the absolute activity of gamma-active nuclides was measured by Symkanych et al. [43]. Based on the data obtained, a map of the distribution of the total gross content of heavy metals and radionuclides was formed, which allowed for the evaluation of the migration process and the identification of regions with a disturbed balance of harmful substances.

According to the data of Lee et al. [50], the monitoring of radioactive pollutants, mainly lying at a depth of 15 cm of the soil surface layer, can be carried out using several radiochemical analytical methods: plasma or laser spectrometry; and scintillation or semiconductor spectrometry. Plasma or laser spectrometry can effectively detect vertical variations in surface contamination only at a depth of about 10 cm because of its minimal penetration depth. Therefore, mobile scintillator spectrometry was proposed to comprehensively characterise the radioactive contamination of decommissioned nuclear facilities. In the study by Lee et al. [50], a mobile in situ scanning system, consisting of a gamma-ray spectrometer, was developed and tested for application in nuclear decommissioning sites. The results demonstrated its potential as an integrated performance-assessment tool for in situ monitoring at nuclear decommissioning sites.

Minimising the pollution of agricultural products is the main direction of the state in ensuring environmental safety and public health, as radionuclides enter the human body

during the consumption of contaminated products. This relationship characterises the trophic chain: radioactive fallout–soil–agricultural plants–farm animals–humans [51].

It is possible to form three main migration flows of radionuclides that fell on the territory of Ukraine (Figure 3) [50–52].

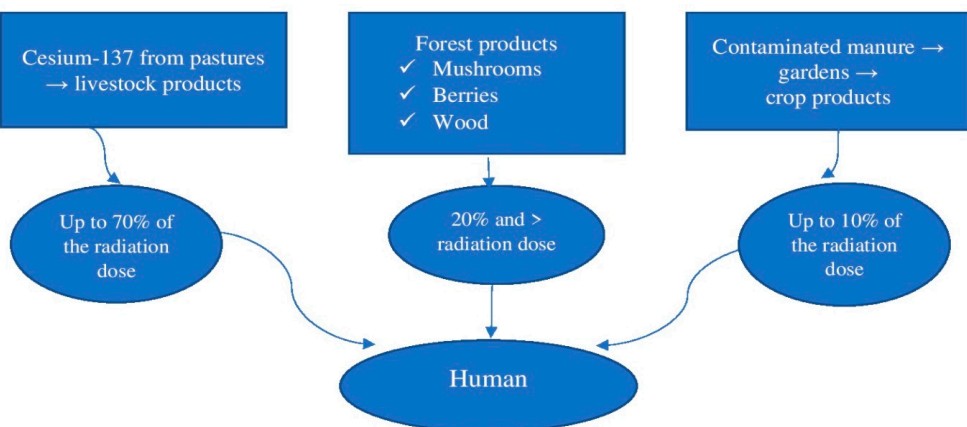

**Figure 3.** Radionuclide migration flows through the trophic chain to the human body.

The set of measures that prevents the entry of radionuclides into agricultural products includes the following [7,53]:

- Natural autorehabilitation (radioactive decay, and fixation and redistribution of radionuclides in the soil);
- Strengthening of biogeochemical barriers to fix radionuclides in soils, reducing the risk of radiation contamination of food;
- Strengthening the radioecological monitoring of soils and agricultural products, radiological control, and compliance with recommendations for agricultural production.

The restoration of radioactive soils is carried out using methods based on such strategies as dry separation, soil washing, flotation separation, thermal desorption, electrokinetic remediation, phytoremediation, etc. The main factors that help to select soil-cleaning methods effectively include soil type, particle size, percentage of fine particles, and radionuclide characteristics [54].

The characteristics and composition of radioactive particles depend on the source of release, and emission scenarios affect the properties of these particles, which is directly essential for the transfer to the environment. Radioactive particles in the bio-environment can come in a variety of physical and chemical forms, ranging from low-molecular-weight particles, colloids, or nanoparticles to pseudocolloids, particles, and fragments. Therefore, information on the types of radionuclides that transform over time is important for assessing the state of contaminated areas and irradiated organisms [55,56]. Radioactive particles can also carry a certain amount of radioactivity and be point sources of radiological danger [57].

For the sorption of radionuclides and heavy metals, various matrices can be used. Basic rock-forming minerals (framework aluminosilicates) are better suited for immobilisation of radionuclides of alkaline and alkaline-earth element groups, as well as halogens, and the use of accessory minerals (phosphates, titanates, and titanium zirconate). As reviewed in our previous studies [14], matrix materials such as phosphates, zirconolites, and sphenes can be recommended for use. In more detail, it is worth dwelling on biosorption methods of ecosystem remediation, which are of increasing interest in applied technologies of radionuclide and heavy metal fixation.

## 4. Biotechnologies for Integrated Fixation of Heavy Metals and Radionuclides: Identification of Advantages and Disadvantages

### 4.1. Soil Bioremediation Methods

The soil rehabilitation process of microbes is carried out using mechanisms such as bioprecipitation, biosorption, bioaccumulation, bio-assimilation, bio-extraction, biodegradation, and biotransformation [58–63]. Some methods for fixing heavy metals and radionuclides are shown in Figure 4. In situ remediation, which involves treating the contaminated site directly in place, can be further subdivided into intrinsic bioremediation and engineered bioremediation. Intrinsic bioremediation occurs naturally without human intervention, while engineered bioremediation involves manipulating the environment to accelerate the degradation of the contaminant [64].

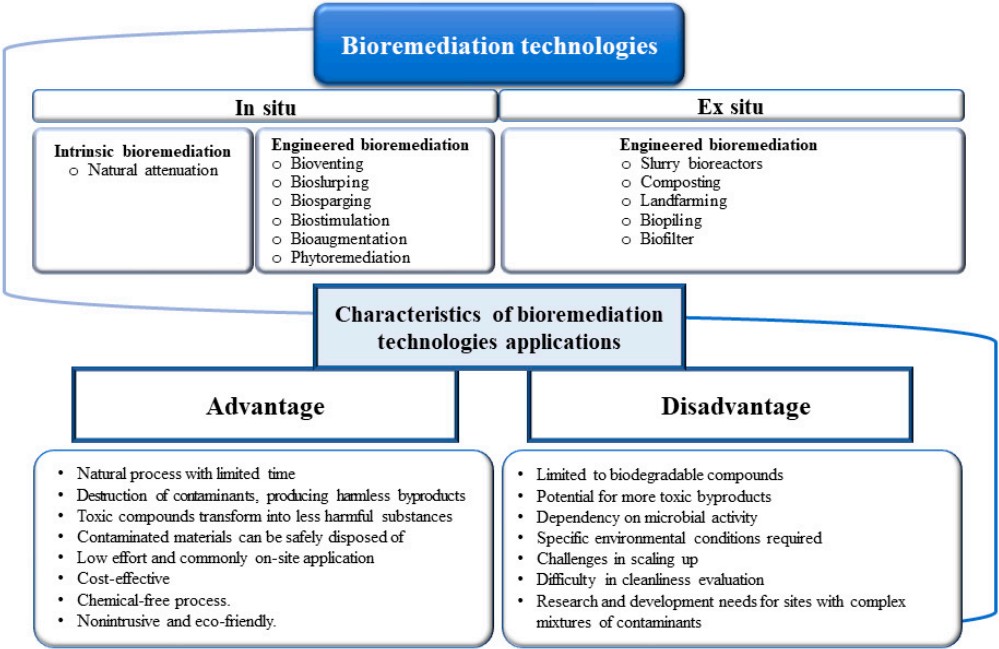

**Figure 4.** Generalisation of remediation methods.

The connection between engineered bioremediation methods and physical and chemical treatment methods is to complement each other. Therefore, stage-by-stage soil remediation is needed, including physical, chemical, and biological treatment methods. Physical and chemical methods can precede biological methods and serve as a preliminary stage. Groudeva et al. [59] investigated the dissolution and removal of contaminants from soil using $Na_2CO_3$ and $NaHCO_3$ solutions, linked to the activity of heterotrophic and basophilic chemo-lithotrophic microorganisms. This activity was intensified by the corresponding changes in environmental factors, such as water, oxygen, and nutrient levels. Furthermore, dissolved-impurities soil leachates were efficiently treated using a nearby natural wetland ecosystem [59].

In the context of mechanical and physicochemical soil remediation methods, the contaminated soil fraction is excavated and then transported to a designated disposal site, where it is stored and treated, incurring additional space requirements and transportation costs [60]. This approach has the disadvantage that it essentially relocates contamination to another location, necessitating ongoing monitoring of the previously contaminated soil and the surrounding environment. Furthermore, during the removal and transport of contaminated soil, there is a risk of spreading contaminated soil and dust particles.

Chemical or physicochemical remediation can be used as a standalone method (when heavy metal concentrations are less than 100 mg/L), but it is more advisable to use it as a preliminary step before biological remediation. The latter approach allows for the

removal of heavy metals from an environment with concentrations significantly lower, but exceeding background levels due to pollution [61].

The chemical and physicochemical methods in the separate application require soil treatment with certain reagents and subsequent leaching with an organic or inorganic solvent, which can lead to deterioration of soil properties, creating an additional factor of destruction of natural soil properties, excluding the possibility of their further use [45].

There are many factors to consider when using physicochemical methods, e.g., pH, temperature, time, nature of the desorbing agent, etc., making the physicochemical method not always suitable, effective, or economically feasible [32]. For example, ion exchange, as a chemical treatment method, can be used to remove various types of metals from the soil but requires the replacement of ion exchange materials and can be expensive [56,63,65].

Compared to organic contaminants, heavy metals and radionuclides in soil cannot be destroyed but must either be converted into a stable form or removed. For this purpose, it is appropriate to use chemical methods to clean soils contaminated with heavy metals and radionuclides, which allow the reaction mixture to be applied directly to the contaminated area, while the topsoil that is being cleaned does not have a significant impact on the functioning of the ecosystem in general [66].

One such approach for the purification of heavy metal-contaminated chernozem soils involves the incorporation of a residual mixture of organic and mineral compost. In this scenario, pollutants are not extracted from the soil; instead, they are temporarily transformed into less readily available forms for plants over a specific duration, typically 4–5 years. However, this method itself does not provide a solution to the problem of removing pollutants from soils but can be combined with biological methods to achieve a positive effect.

The biological soil remediation of heavy metals and radionuclides is achieved through biotransformation. Microorganisms, such as bacteria, fungi, and microscopic algae that reside in the soil, are effective biotic entities that are capable of efficiently absorbing or transforming heavy metal and radionuclide compounds [67].

Heavy metals that penetrate living cells exhibit their toxic effects primarily in the form of ions. However, if heavy metals and radionuclides are transformed into bound forms through various means, they lose their toxic properties [68]. Consequently, heavy metals deposited in the cell wall in a crystalline or poorly soluble compound form become non-toxic to microorganisms but are eventually removed from the environment as a result of biological remediation.

The mechanisms through which microorganisms interact most frequently with heavy metals include biosorption (the sorption of metals on cell surfaces through physicochemical mechanisms), bioleaching (the mobilisation of heavy metals through the excretion of organic acids or methylation), biomineralization (the immobilisation of heavy metals through the formation of insoluble sulphides or polymeric complexes), bioaccumulation (intracellular accumulation), and enzyme transformation catalysis (oxidation-reduction reactions) (Figure 4) [69,70].

Biological methods of soil remediation offer partial solutions to challenges in this field. From an economic point of view, they provide benefits by avoiding the need for significant one-time investments. The associated costs can be spread over several years. These methods also eliminate the requirement for mandatory soil excavation and can be applied to larger areas. Furthermore, they avoid the introduction of specific harmful chemical mixtures, solutions, or reagents into the soil, thus preventing secondary pollution [71,72]. The general disadvantages of biological methods are their delayed effectiveness; long duration; and dependence on climatic conditions, including the rate of development of bioremediation organisms and biotransformations carried out by microorganisms in climatic conditions with variable temperature and humidity throughout the year [73,74].

Table 3 presents a classification of soil bioremediation methods. The approach chosen may vary depending on the concentration and type of target metals. It is also essential to consider an ecosystem-based approach within the context of interconnectedness because

the soil environment interacts with water resources and the atmosphere, influenced by the biochemical activities of organisms in the natural components of the ecosystem [75–77].

**Table 3.** Classification of soil bioremediation methods.

| Method | Brief Definition | Process Features Considering Their Limitations | References |
|---|---|---|---|
| Biomineralisation | Deposition of heavy metals as insoluble compounds. It includes two primary methods: microbiological carbonate precipitation and enzymatic carbonate precipitation. | It is considered an environmentally friendly bioremediation method that is not less effective than chemical methods. However, limitations related to microorganism strains, pollutant concentrations, and soil properties must be taken into account. Further research on soils treated with biomineralization, the solidification and stabilisation (S/S) of toxicants, is necessary to understand the patterns of strength change in polluted soils treated with biomineralization. Additionally, it is important to investigate changes in the rate of heavy metal fixation and the mechanical properties of contaminated soil. | [77–83] |
| Biosorption | This is a physicochemical and metabolically independent process that relies on various mechanisms, including absorption, adsorption, ion exchange, surface complexation, and precipitation. | Advantages include low cost and significantly higher efficiency in removing metals from diluted solutions. Heavy metal adsorption and removal can be performed using biomass, which can generate income for businesses that do not use biomass, such as organic waste. Various environmental parameters, such as temperature, metal type and concentration, metal oxidation state, microbe type, metal removal method, and biosorbent concentration, can influence the ability of microorganisms to bind metals. This may have a negative impact on biosorption efficiency. | [84–92] |
| Bioprecipitation | In the process of bioprecipitation, the formed metabolites react with metals present in the groundwater, resulting in the precipitation of metals, i.e., the transformation of metals from the aqueous phase to the solid phase. | Bioprecipitation is more effective in treating wastewater than soils; however, the profitability of recycling or selling recovered metals can vary depending on the investments in infrastructure of the investments in infrastructure of a company. It is recommended to use it in conjunction with other biological methods. | [78,93–98] |
| Bioaccumulation | Active uptake of heavy metals into cells involves the binding of toxic metals or chemical compounds inside the cellular structure. | This method not only is cost-effective but also helps minimise the environmental impact of pollution. Metal bioaccumulation is particularly useful as an impact indicator, as metals are not metabolised. Metal ions initially attach to the cell surface and are later transported into the cell. This process can lead to a temporary reduction in metal ion concentration. However, it can be utilised to synthesise metal-rich nanoparticles, provided that the processing is performed in specialised bioreactors rather than in situ. | [85,99–105] |
| Biotransformation | Breakdown of heavy metal compounds into less toxic forms or their conversion into less toxic forms (associated with biodegradation). | Photoautotrophic microbes are capable of biotransforming heavy metals into relatively biologically inaccessible and insoluble metal sulphides. By characterising the role of sulphur assimilation pathways in the biotransformation of heavy metals, we can develop more effective processes for heavy metal bioremediation. The use of additional sulphate nutrition can enhance the rate of biotransformation in aerobic microbes. | [78,85,106–111] |

Bioremediation-based processes can be considered a promising area based on the transformation of heavy metals and radionuclides into a less dangerous state and, at the same time, provide sustainable restoration of the environment. Thus, as part of the study of transformations of metals such as Pb, Zn, and Cd by Thakare et al., a number of regularities were identified. Metals cannot be decomposed by microorganisms involved in contaminated soil rehabilitation, but they can be changed from one oxidised form to another, allowing them to become fixed in insoluble form and be removed from biogeochemical cycles of migration in the environment [112].

Heavy-metal ions and radionuclides can usually be adsorbed by functional groups such as carbonyl, carboxyl, sulfhydryl, phosphate, sulphate, amino, and hydroxyl groups on the bacterial surface. The ability of bacteria to absorb heavy-metal ions generally varies from 1 mg/g to 500 mg/g [113]. Extracellular polymer substances consisting of proteins, lipids, nucleic acids, and complex carbohydrates play an important role in the adsorption of heavy-metal ions. These substances on the surface of the bacterial cell can prevent heavy metal toxicity and penetration into the inner cell region [112]. When studying the effect of metals on soil biological properties, it is feasible to use a set of methods, such as microbial biomass, C and N mineralisation, respiration, and enzyme activity, that will allow for a complete evaluation of this interaction [114].

In general, the following types of microorganisms are used in the bioremediation methodology [78]: *Bacillus* sp., *Lysinibacillus* sp., *Rhodococcus* sp., *Ascomycota*, *Basidiomycota*, *Perenniporia subtephropora*, *Daldinia starbaeckii*, *Phanerochaete concrescens*, etc.

*4.2. Biosorption Technologies and Their Aspects of Realisation*

Today, biosorption has been accepted as an environmentally friendly alternative green technology for the removal of various human-made pollutants, with the help of microbes such as bacteria, fungi, algae, and yeast. Pollutants are substances that do not decompose, are relatively unyielding, are insoluble in water, are impervious to microbial cells, and are harmful to lower and higher classes of living organisms. Desorbing eluents can be used to remove adsorbed pollutants, and biosorbent regeneration can be carried out by chemical, thermal, or electrochemical methods [115].

Fundamental to understanding the biosorption process is knowledge of the mechanism of the process. Based on cellular metabolism, biosorption mechanisms can be classified into independent and dependent mechanisms. Based on the location of biosorption, the following are distinguished [116–121]: (i) intracellular accumulation, (ii) extracellular accumulation and deposition, and (iii) cell surface sorption and deposition. The mechanisms belonging to the first two groups depend on metabolism and are caused by the processes of complexation, precipitation, and ion exchange; and the last group of mechanisms are also adsorption (physical and chemisorption).

The process of adsorption involves the attraction of other dissolved particles to the surface of a solid substance (adsorbent), primarily through adhesion, electrostatic attraction, and ion exchange. The adsorbent "fixes" all contaminants in its structure and thus purifies the sample [116].

An integrated approach is necessary for the restoration, regeneration, revegetation, and management of areas with a high level of anthropogenic loads, such as areas contaminated with heavy metals and radionuclides. Methods can be applied effectively for soil restoration, including some green activities, such as phytoremediation, and an appropriate soil cleanup process can be established. The enhancement of phytoremediation takes place through organic additives, namely agricultural waste and pretreated sewage sludge, biochar, humic substances, plant extracts, exudates, etc. [117].

At the same time, the commercialisation of biosorption technologies is hindered by technical problems associated with the operation and regeneration of native biosorbents. This problem is partially solved by immobilising microorganisms in a solid inert carrier, such as biochar, zeolites, and vermiculite, or by including them in an alginate gel. In this case, it becomes possible to apply a dynamic sorption process, the so-called "column

variant" [118–121]. However, the sorption capacity of the biosorbent decreases significantly in comparison to static biosorption. Furthermore, there are still problems with biosorbent regeneration and replacement in the event of complete depletion.

The key advantages and disadvantages of biosorption technologies are shown in Figure 5. Biosorption is well suited for use in large areas of contaminated soil where other remediation methods are not economically feasible or difficult to implement practically and where soil productivity can be restored over long periods of time. It can be combined with other technologies, such as phytoremediation, for the final closure of the site with vegetation. In emergency situations or military action that involves the release of high concentrations of pollutants into the ecosystem, it is initially necessary to use physicochemical methods to quickly stop vertical and horizontal migration into natural components. Biosorption technology has some limitations that should be considered before choosing it for the remediation of areas contaminated with heavy metals and radionuclides: the prolonged duration of territorial restoration and the fixation and transformation of pollutants into less toxic forms have a long-term positive effect, but there is a potential risk of contamination through the food chain.

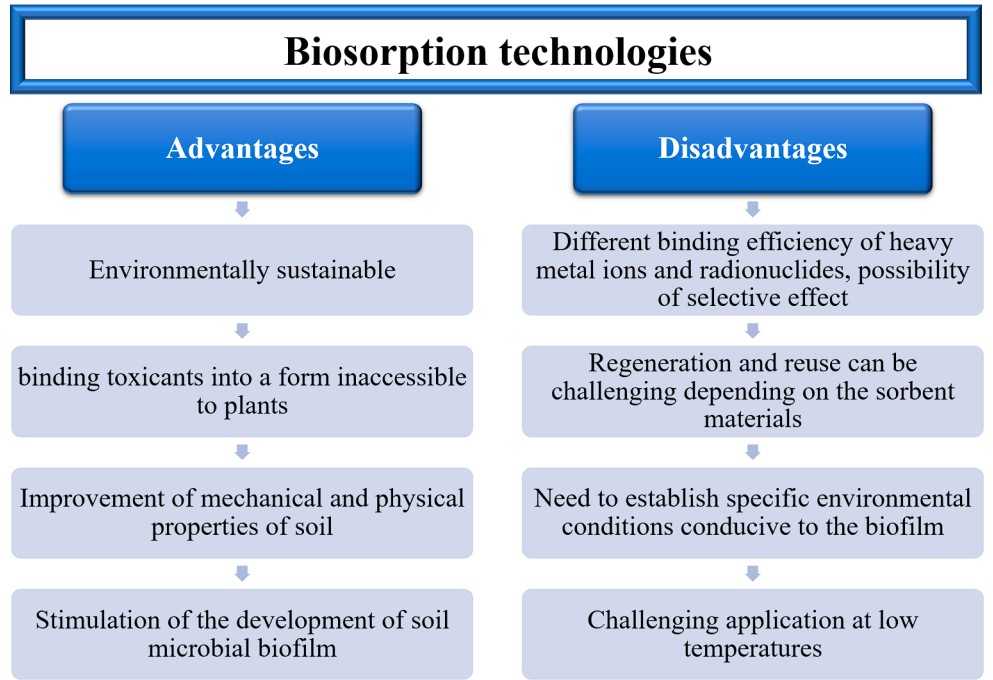

**Figure 5.** Characteristics of biosorption technologies.

A biostimulation approach is used to improve biosorption processes. It includes stimulating the growth of microorganisms in a contaminated soil area to introduce pH-correcting substances, nutrients, surfactants, and oxygen [122], which requires further research.

However, it should be noted that there is a lack of information on the synergistic or inhibitory effect on the sorption processes of metal ions in multicomponent solutions with different ionic strengths, effective methods of immobilisation of microorganisms for the implementation of flow biosorption processes, selectivity and ways to increase it in the concentration of heavy metals, etc. This indicates the need to create a research algorithm for the study of biosorption processes using microorganisms.

Thus, to date, the following directions are relevant [60,123–128]:

- Studies of microorganisms of different physiological groups (including the use of genetically modified strains) on the ability to sorb and transform soluble forms of heavy metals and radioactive elements into insoluble ones;

- Bacterial reduction processes of technetium, chromium, and uranium when used as final electron acceptors in bacterial energy metabolism for the purpose of their detoxification in systems with neutral, acidic, and alkaline pH values;
- Determining the products of bacterial transformation of radionuclides and heavy metals formed under different conditions;
- Possibilities of reducing the toxic effects of heavy metals and radionuclides on soil microorganisms;
- Development of nanobioremediation technology.

## 5. Possibility of Using Phosphogypsum for Soil Bioremediation

The use of phosphogypsum is associated with challenges that have gained increasing importance [129], as shown in Figure 6. Furthermore, it should be noted that phosphogypsum may be contaminated with radionuclides [130]. According to EPA data, phosphogypsum contains significant quantities of uranium and its decay products, such as radium-226, attributed to its presence in phosphate ores. The concentration of uranium in phosphate ores identified in the United States varies within the range of 0.26 to 3.7 Bq/g (7 to 100 pCi/g) [131]. However, various raw materials are used in different countries and regions globally; consequently, not all phosphogypsum exhibits elevated levels of radioactivity [129,132].

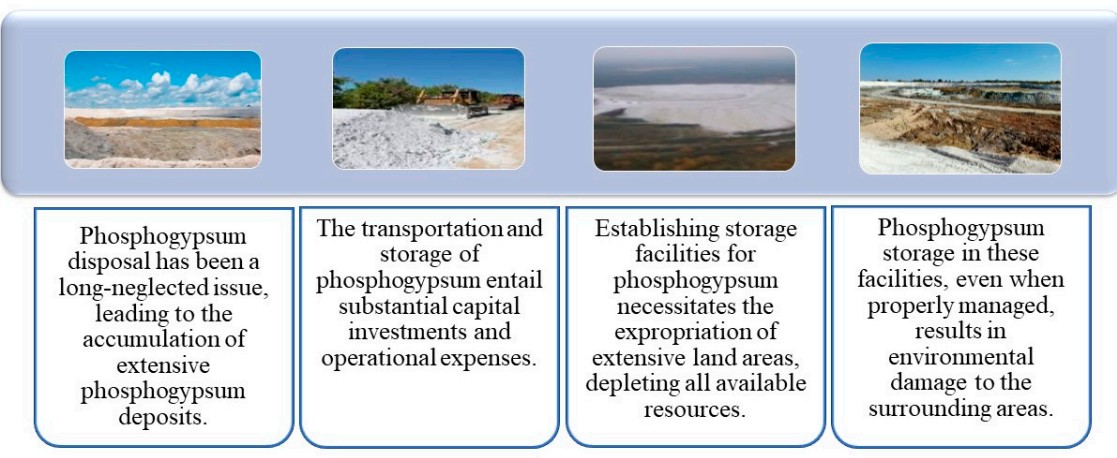

**Figure 6.** Accumulation of phosphogypsum in the environment.

In order to address the development of environmentally friendly technologies for the use of phosphogypsum within the context of the bionics concept that integrates biological methods and structures for engineering solutions and technological approaches, it is necessary to improve the technical solutions and technologies for phosphogypsum utilisation in potential soil applications. A crucial element involves precise control over the composition of the soil solution through in situ synthesis of essential compounds directly within the soil. Given the present issue of soil degradation, there is a pressing need to actively explore novel soil management strategies. Furthermore, the effective resolution of this problem requires the availability of suitable design tools [18].

We emphasise the use of phosphogypsum, which does not have significant radioactive contamination, in bioprocesses. Therefore, Figure 7 shows the main elements of phosphogypsum that positively affect soil properties [133,134]. Furthermore, phosphogypsum has an impact on the growth of microorganisms, which has been confirmed by several studies [135–139]. Through the regulation of soil moisture, it is possible to significantly reduce the leaching of unproductive substances and address issues related to the hydromorphic regime of the soil, including the degradation of organic matter and the reduction of sulphate to sulphides. Moisture control also enhances the protective effect of the geochemical

barrier "soil–rhizosphere", effectively retaining harmful compounds within soil solution, particularly for Pb, Cd, and Sr [140].

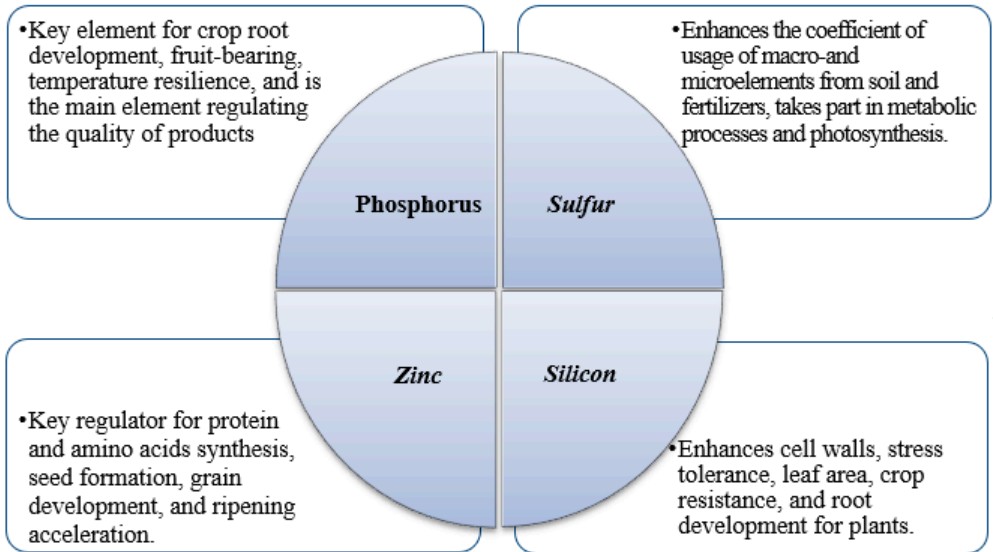

**Figure 7.** Beneficial elements in the composition of phosphogypsum (case study: phosphogypsum dump in Sumy region, Ukraine).

Figure 8 shows the distribution of countries according to their publication activity in the phosphogypsum research documented in the Scopus database. Distribution of countries by publication activity in the field of phosphogypsum research according to the Scopus database.

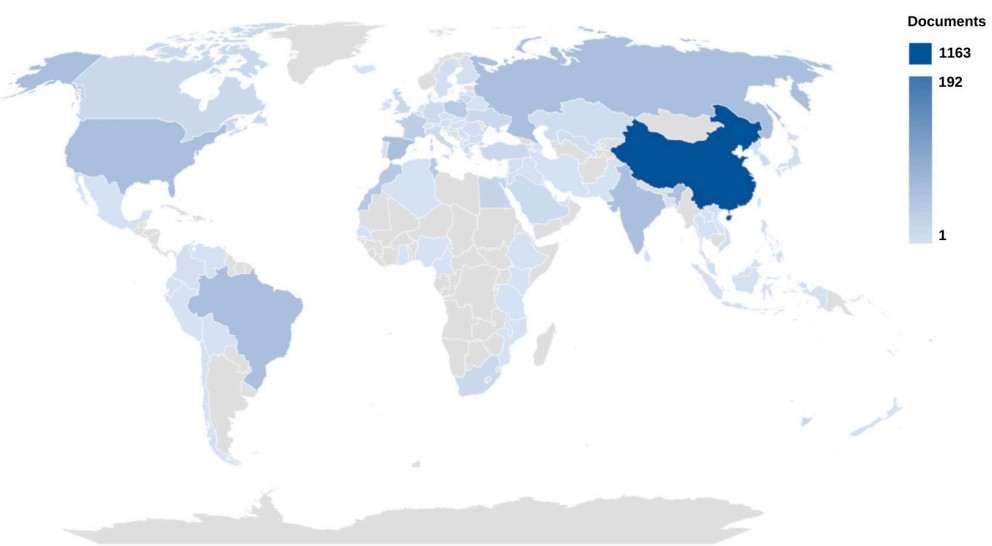

**Figure 8.** Distribution of countries by publication activity in the field of phosphogypsum research according to the Scopus database.

Tables 4 and 5 show a comparative analysis of the concentrations of elements in phosphogypsum from different countries.

Table 4 shows that the main components of calcium and sulphur oxides fluctuate in significant intervals in phosphogypsum samples from different regions of the world, with calcium in the range of 17.7–45.9 wt% and sulphur in the range of 17–51.4 wt%, respectively. At the same time, components such as iron, potassium, aluminium, magnesium, and manganese also have a significant difference in the amount of content in phosphogypsum

from different locations of generation in the world. This is due to the technological process of production, but the special influence on changes in the content of trace elements is influenced by the raw materials used (phosphates and apatites).

In terms of the content of radionuclide isotopes, the data vary significantly depending on the region of phosphogypsum deposition. A review of previous studies [130,141,142] showed that radioactivity varies according to the type of phosphate ore and is mainly caused by the decay series U-238 and Th-232. Since U-235 is not as common in nature as U-238, the radiation of this decay series is not considered a threat [143]. However, the information about harmful impurities in phosphogypsum related to its environmental impact is not yet fully understood, which requires scientific evaluation in the future and the expansion of research in this area [144].

Therefore, it is worth concluding that Ukrainian phosphogypsum (in particular, from the Sumy region, since its samples were studied) is the most environmentally acceptable for bioremediation processes (Tables 4 and 5):

- Heavy metals (e.g., As, Pb, and Cr) have lower concentrations in phosphogypsum from the Sumy region than in phosphogypsum from China, Spain, the USA, and Brazil;
- Some rare earth elements (such as La, Ce, Pr, and Y) are represented in phosphogypsum from the Sumy region (Ukraine) and less represented in phosphogypsum from other regions of the world.

However, this conclusion requires several further studies on the testing of Sumy phosphogypsum on different types of soil in bioremediation practice.

**Table 4.** Concentrations of major elements in phosphogypsum.

| wt.% | Ukraine [a] | China [b] | United States [c] | Spain [d] | Brazil [e] | India [f] | Morocco [g] | Poland [h] | Tunisia [i] | France [j] | Greece [k] |
|---|---|---|---|---|---|---|---|---|---|---|---|
| CaO | 22.9–31.4 | 31.6–43.3 | 22.7–39.4 | 17.7–32.6 | 31–36 | 30.9–38.9 | 32.2–35 | 29.6–42.7 | 30.7–37.2 | 31.3–33.4 | 34.30 |
| $SO_3$ | 29.8–36 | 34–49 | 22.9–51.9 | 30.7–46 | 44.5 | 44.2–52.9 | 17–45.1 | 42.1–56.5 | 37.5–47 | n.m. | 41.50 |
| $SiO_2$ | 13.1–24.7 | 3.6–15.3 | 3.2–51.3 | n.m. | 0.8 | 0.5–4.3 | 0.3–9.7 | 0.4–1.8 | 1.0–3.8 | 0.6–1.5 | n.m. |
| $Al_2O_3$ | 0.96–2.52 | 0.08–2.59 | 0.069–1.14 | n.m. | 0.11–0.2 | 0.1–0.77 | 0.13–0.77 | 0.18–1.7 | 0.04–0.11 | 0.11–0.31 | n.m. |
| $P_2O_5$ | 0.63–0.79 | 0.68–1.82 | 0.5–3.8 | 0.49–1.18 | 0.07–1.29 | 0.82–1.04 | 0.59–1.62 | 1.5 | 0.8–1.69 | 0.36–0.69 | n.m. |
| $Fe_2O_3$ | 0.41–0.94 | 0.05–1.95 | 0,13–1.15 | n.m. | 0.25–0.77 | 0.1–0.56 | 0.15–0.83 | 0.06–0.20 | 0.03–0.13 | n.m. | 0.84 |
| $K_2O$ | 0.1–0.32 | 0.17–0.33 | 0.02–0.9 | 0.02 | 0.04 | 0.03 | 0.05–0.4 | n.m. | 0.01–0.03 | n.m. | n.m. |
| $TiO_2$ | 0.05–0.17 | 0.04–0.27 | 0.03–0.46 | n.m. | 0.18–0.52 | 0.02–0.05 | 0.01–0.03 | n.m. | n.m. | n.m. | n.m. |
| $Na_2O$ | 0.02–0.07 | 0.05 | 0.11–1.42 | 0.02 | 0.02–0.09 | 0.03–0.11 | 0.14–0.55 | n.m. | 0.05–0.29 | 0.02–0.19 | n.m. |
| MnO | 0.01 | 0.08–0.18 | 0.06–0.07 | n.m. | 0.004–0.017 | n.m. | 0.01 | n.m. | n.m. | 0.0002–0.0004 | n.m. |
| MgO | 0.01 | 0.01–0.23 | 0.03–0.13 | n.m. | 0.02–0.76 | 0.02–0.56 | 0.21–0.54 | n.m. | 0.01–0.07 | n.m. | 0.13 |

[a] Ukraine (author's results); [b] China [145–149]; [c] United States [150–152]; [d] Spain [153,154]; [e] Brazil [155–157]; [f] India [158–160]; [g] Morocco [161–165]; [h] Poland [166–170]; [i] Tunisia [171–174]; [j] France [175]; [k] Greece [176,177]; n.m., not measured.

The selective removal of $Na^+$ and $Cl^-$ from soil, without affecting other macroelement ions, is an integral aspect of the scientific and technical field known as biogeosystem engineering. Biogeosystem engineering deals with engineering solutions and technologies, unprecedented in nature, aimed at managing the cycling of biogeochemical substances in gaseous, liquid, and solid phases. Its primary focus is the ecologically safe use of substances in soils, the improvement of resources and food products, and the solution of the production and environmental challenges in the noosphere through a unified technological cycle based on the principle of natural consistency. In the context of ensuring a quality environment for healthy living, the issue of phosphogypsum involves considering methods for its neutralisation as a more environmentally friendly alternative to the disposal in storage facilities [18,178]. However, for the reclamation of saline soils, neutralising phosphogypsum should be avoided, as its residual acids enhance the solubility of calcium compounds in the soil, promoting sodium displacement by calcium. Therefore, the supply of phosphogypsum to consumers in reusable containers for soil application appears to be a rational solution.

Mixing phosphogypsum with ash from a power plant appears promising for optimising the use of by-products [179]. The lower the coal quality, the higher the CaO content in the ash, leading to a higher level of phosphogypsum neutralisation. Simultaneously, both materials can be recirculated in the soil.

**Table 5.** Concentrations of trace elements in phosphogypsum.

| ppm | Ukraine [a] | China [b] | United States [c] | Spain [d] | Brazil [e] | India [f] | Morocco [g] | Poland [h] | Tunisia [i] | France [j] | Greece [k] |
|---|---|---|---|---|---|---|---|---|---|---|---|
| Cu | 3.6–7.0 | 27.6 | 2.5–35.1 | 2.5–11 | 6.3–9 | n.m. | 1.5–2.9 | 3.39 | 6–9.6 | 5.4–17.5 | 13 |
| As | <4.96 | 7.15 | 0.77–20.1 | 0.6–8.56 | n.m. | n.m. | 1.84–1.94 | 8.05 | 1 | n.m. | 0.61–17 |
| Pb | 4.6–4.7 | 28.15 | 2.06–11.4 | 1.99–10.8 | 7.2–31 | 0.07 | 0.17–1.7 | 10.4 | 0.9 | 1.68–4.57 | 11 |
| Zn | 3.2–19.7 | 37.5 | 1.19–32.1 | 1.92–13.1 | 4.4–85.1 | n.m. | 3–28 | n.m. | 9–137 | n.m. | 12–123 |
| Cr | 4.6–11.9 | 37 | 1.69–20.2 | 3.59–20.3 | 11.1–14.7 | 2.73 | 5.85–11 | 5.9 | 6–13 | n.m. | 15.8–153 |
| Ni | 1.4–1.7 | 16.6 | 0.21–17.79 | 0.87–2.67 | 5.4–11 | 14.48 | 1.2–300 | 3.6 | 0.94–4.1 | n.m. | 21 |
| Cd | 1.19–6.36 | 0.48 | 0.28–10.8 | 1.39–2.83 | <0.1 | n.m. | 0.8–7.38 | 1.7 | 8–17.7 | 1.2–2.1 | 0.98–6.67 |
| V | 1.6–2.2 | 27.5 | 0.38–10.7 | 2.9–12.8 | 6.9–9.2 | n.m. | 1.94–5 | n.m. | 2–3 | 1.43–3.91 | n.m. |
| Ga | 0.49–0.78 | n.m. | n.m. | n.m. | 9–10.4 | n.m. | n.m. | n.m. | 0.87 | n.m. | n.m. |
| Sr | 981 | n.m. | 1.05–899 | 360–596 | 4884.9–6179.1 | n.m. | 530–778 | n.m. | n.m. | 813.2–1275 | 172–470 |
| Ba | 20.5–27.2 | 215 | 30.3–88.9 | 37 | 767.1–6104 | n.m. | 23–63.3 | n.m. | 10 | 92.36–215.6 | 38.3–331 |
| Y | 197.2–148.8 | 74 | 43.36 | 106–142 | 90–105.3 | n.m. | 127 | n.m. | 53.2 | 34.65–100.7 | n.m. |
| La | 195.3–137.1 | 36.5–46 | 36.38 | n.m. | 921.1–1969 | n.m. | 60.7 | 40 | 46.3 | 12.96–43.35 | 24.9–30.5 |
| Ce | 282.1–200 | 30.6–32 | 63.84 | 19.5–81.2 | 2109.1–3547 | n.m. | 39 | 53 | 74.4 | 6.53–18.72 | 19.2–60.7 |
| Pr | 46.7–33.4 | 5 | 5.01 | n.m. | 256.1–276.2 | n.m. | 11 | 8 | n.m. | 1.9–6.9 | n.m. |
| Eu | 0.98 | n.m. | 1.4 | n.m. | 23.7–25.9 | n.m. | 2.48 | 2 | n.m. | 0.49–1.7 | 0.85–1.08 |
| Cs | 0.38 | n.m. | n.m. | n.m. | <0.1 | n.m. | n.m. | n.m. | 0.05 | n.m. | 0.09–4.82 |
| Th | 3.3–5.8 | n.m. | n.m. | 1.1 | 67.2–81 | n.m. | 3.04–3.27 | n.m. | 0.74 | 0.22–1.39 | 0.59–10.1 |

[a] Ukraine (author's results); [b] China [145–149]; [c] United States [150–152]; [d] Spain [153,154]; [e] Brazil [155–157]; [f] India [158–160]; [g] Morocco [161–165]; [h] Poland [166–170]; [i] Tunisia [171–174]; [j] France [175]; [k] Greece [176,177]; n.m., not measured.

In our previous study, Chernysh et al. [18], the introduction of phosphogypsum into the process of anaerobic fermentation of sustainable feedstock (sewage sludge, etc.) leads to the introduction of additional macroanalogues into the organo-mineral structure of digestate. It should be noted that the introduction of phosphorus and calcium compounds contained in phosphogypsum intensified the process of fixation of heavy metals and radionuclides in the sludge. As a result, calcium and potassium hydrogen phosphate compounds, which have the ability to adsorb radionuclides, were found in the mineral composition of the digestate [18].

The factors that influence the migration of radionuclides into the ecosystem and the impact of the organo-mineral complex on the fixation of heavy metals and radionuclides in soils are described in Figure 9.

Thus, the uptake of radionuclides by plants and their accumulation by chemicals in crop fields are largely dependent on the amount of their chemical analogues in the environment. An increase in the exchange capacity usually leads to an increase in the adsorption strength of radionuclide traces. Therefore, the accumulation of 137Cs by plants in most cases is inversely proportional to the absorption capacity of the soil and the amount of exchangeable K in it, and for 90Sr [3]. The uptake of 90Sr and 137Cs by plants decreases with an increase in the content of calcium and potassium in the soil or growing medium [18].

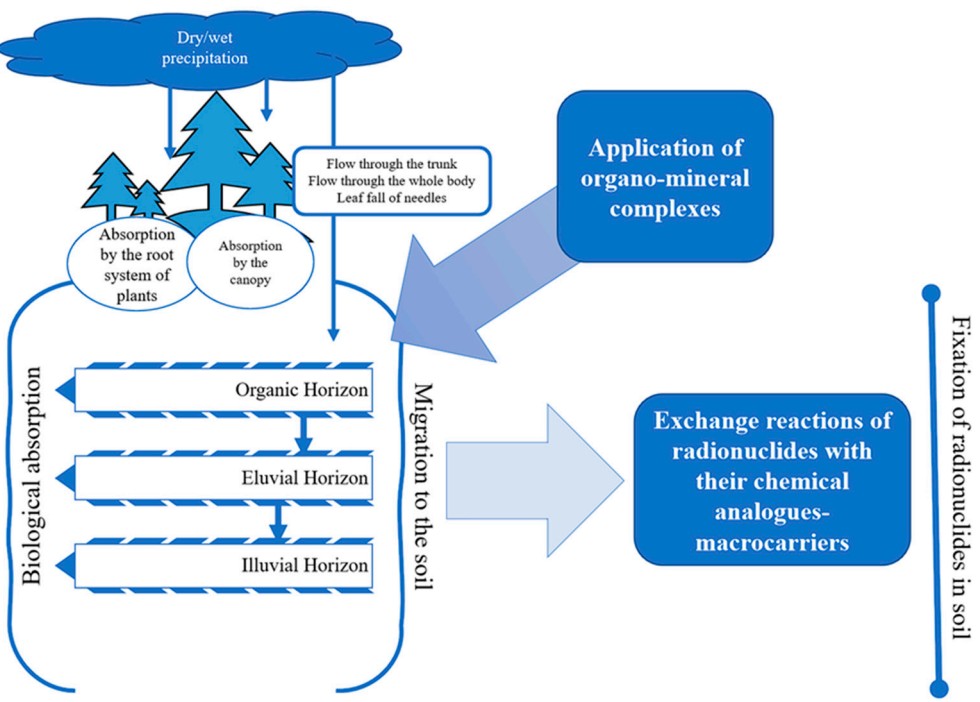

**Figure 9.** Influence of the organic–mineral complex on the fixation of heavy metals and radionuclides in soils.

## 6. Conclusions

A review of studies of heavy metal content was conducted in radioactively contaminated areas. In particular, the sources of heavy metals and radionuclides in the soil and their impact on ecosystem services with inclusion in food chains were identified. This article discusses the important issue that is the remediation of soils contaminated with heavy metals and radionuclides, especially if these toxicants are present simultaneously in contaminated areas. Therefore, remediation methods should take into account the specificity of both of them.

The advantages and disadvantages of immobilising heavy metals and radionuclides are identified using biosorption methods. Technical problems associated with the use and regeneration of local biosorbents have hindered the commercialisation of biosorption technologies. The immobilisation of biomass on solid inert carriers (e.g., biochar, zeolite, and vermiculite) can partially solve this problem. However, the issue of regeneration and replacement of the biosorbent in case of its complete exhaustion arises. In addition, the directions for the use of phosphogypsum as a sorption carrier for soil bioremediation were determined. It is necessary to take into account the neutralisation of phosphogypsum in this field as a promising one, which requires further research within the framework of the development of the biogeosystem approach. It should be noted that the post-war restoration of the contaminated territories of Ukraine is a complex and strategic task within the framework of the global issue of food security.

**Author Contributions:** Conceptualisation, Y.C.; validation, H.R.; formal analysis, P.S., V.C. and M.S.; investigation, V.C. and P.S.; writing—original draft preparation, Y.C.; writing—review and editing, I.A., V.C., P.S., O.Y., L.P. and H.R.; visualization, I.A., P.S. and V.C.; funding acquisition, Y.C. and H.R.; supervision, H.R. All authors have read and agreed to the published version of the manuscript.

**Funding:** This project, "Phosphogypsum as a mineral resource for bioprocesses", received funding through the MSCA4Ukraine project, which is funded by the European Union (Y.Ch.). This research project was carried out as planned research projects of the Department of Ecology and Environmental Protection Technologies of Sumy State University, related to the topics "Assessment of the technogenic load of the region with changes in industrial infrastructure" according to the scientific and technical

programme of the Ministry of Education and Science of Ukraine (state registration no 0121U114478). Finally, we would like to acknowledge Technology Agency of the Czech Republic (Grant Number TH79020003; support for this project is offered under the coordination of the ERA-MIN3 action, which received funding from the European Union under the Horizon 2020 Programme (European Commission Grant Agreement No. 101003575).

**Institutional Review Board Statement:** Not applicable.

**Informed Consent Statement:** Not applicable.

**Data Availability Statement:** The data sets generated during and/or analysed during the current study are available from the corresponding author on request.

**Conflicts of Interest:** The authors declare that they have no competing interests.

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
