# Peer review of "Soil Contamination by Heavy Metals and Radionuclides and Related Bioremediation Techniques: A Review"

_soilsystems, doi:10.3390/soilsystems8020036_

Round 1

Reviewer 1 Report (Previous Reviewer 4)

Comments and Suggestions for Authors

The article has been revised to improve the logic of the line, writing format and language expression. However, there are still small problems that need further revision and improvement, as follows:

1.Line 46: Is the citation of the literature in the text formatted correctly, please refer to the authors' guidelines to bring it in line with the requirements of the journal. In addition, there are a number of similar citations, so if there is a problem, please check and correct yourself.

2.Line 114: "Review of studies of heavy metal content in radioactively contaminated territories." The sentence focuses on studies of heavy metal content in radioactively contaminated areas. However, according to the later text, radioactive contamination is also the focus of the study. Therefore, is the expression there not accurate enough?

3.Line 345: Is there a problem with the transitive relationship of the sentence?

Comments on the Quality of English Language

There are many issues with the improper use of coniunctions in sentences irnthe article, resulting in unclear logical relationships between sentences. lt isrecommended that you reorganize and adjust the sentence structure, and tryto use concise and accurate language to make the logic of the sentencesclearer and more coherent.

Author Response

The article has been revised to improve the logic of the line, writing format and language expression. However, there are still small problems that need further revision and improvement, as follows:

1.Line 46: Is the citation of the literature in the text formatted correctly, please refer to the authors' guidelines to bring it in line with the requirements of the journal. In addition, there are a number of similar citations, so if there is a problem, please check and correct yourself.

Response to comment

In this paragraph, the citation is correct. In the text, the reference numbers are given in square brackets [ ] according to the requirements of the journal, and surnames were added for clarity of references. But thank you for pointing this out. We have went over the whole manuscript again and added throughout the text.

In research by Morooka et al. [4] are presented areas affected by nuclear power plant (NPP) disasters. Thus, 31 radioactive particles from surface soils were detected in an area 3.9 km northwest of the Fukushima-1 NPP. 134 + 137Cs had the highest activity ever recorded for Fukushima-1 NPP (6.1×105 and 2.5×106 Bq per particle after decay correction until March 2011). Taking into account their large size (120 μm), the impact of these particles on human health will be minimal, including radiation during static skin contact [4].

2.Line 114: "Review of studies of heavy metal content in radioactively contaminated territories." The sentence focuses on studies of heavy metal content in radioactively contaminated areas. However, according to the later text, radioactive contamination is also the focus of the study. Therefore, is the expression there not accurate enough?

Response to comment

Thank you for the comment.

A clarification was made, highlighted in yellow in the text of the manuscript:

Review of the state of ecosystems contaminated with heavy metals and radionuclides.

3.Line 345: Is there a problem with the transitive relationship of the sentence?

 Response to comment

Thank you for the comment. We have rephrased it in the text.

The general disadvantages of biological methods are their delayed effectiveness, long duration, and dependence on climatic conditions, including the rate of development of bioremediation organisms and biotransformation carried out by microorganisms in climatic conditions with variable temperature and humidity throughout the year [73, 74].

Comments on the Quality of English Language

There are many issues with the improper use of coniunctions in sentences irnthe article, resulting in unclear logical relationships between sentences. lt isrecommended that you reorganize and adjust the sentence structure, and tryto use concise and accurate language to make the logic of the sentencesclearer and more coherent.

Response to comment

Thank you for this comment. We have went over the manuscript to eliminate any remaining issues. In addition, we had the manuscript submitted for language proofreading.

Dear Reviewer,

We would like to thank you again for your time and dedication to improve our manuscript.

We have followed all your recommendations and improved accordingly.

Thank you.

With regards,

Authors

Reviewer 2 Report (Previous Reviewer 5)

Comments and Suggestions for Authors

My comments have been properly addressed. 

Author Response

Dear Reviewer,

We want to thank you again for your time and dedication to improving our manuscript.

With regards,

Authors

This manuscript is a resubmission of an earlier submission. The following is a list of the peer review reports and author responses from that submission.

Round 1

Reviewer 1 Report

Comments and Suggestions for Authors

Good reviev article. maybe you should concider adding info of potential phosphgypsum management.

Author Response

Responses to reviewer comments 1

Good reviev article. maybe you should concider adding info of potential phosphgypsum management.

Response

Thank you very much for your opinion. We have taken your comments into account and made the necessary additions to Section 3.3 (highlighted in green).

Dear Reviewer,

Thank you again for reviewing our manuscript and we appreciate your feedback and time dedicated to make our manuscript better.

Reviewer 2 Report

Comments and Suggestions for Authors

This study focuses on the interaction and intricate influence of heavy metals and radionuclides. It proposes potential issues in the rehabilitation processes of contaminated ecosystems, offering valuable insights for the formulation of environmental pollution control strategies. The paper is generally well-written; however, there are a few minor issues that need attention. I recommend considering acceptance after addressing the following:

1. Modify some minor language errors.

2. Ensure uniformity in the reference format, particularly in lines 166, 196, 260, and so on.

Comments on the Quality of English Language

This study focuses on the interaction and intricate influence of heavy metals and radionuclides. It proposes potential issues in the rehabilitation processes of contaminated ecosystems, offering valuable insights for the formulation of environmental pollution control strategies. The paper is generally well-written; however, there are a few minor issues that need attention. I recommend considering acceptance after addressing the following:

1. Modify some minor language errors.

2. Ensure uniformity in the reference format, particularly in lines 166, 196, 260, and so on.

Author Response

Responses to reviewer comments 2

This study focuses on the interaction and intricate influence of heavy metals and radionuclides. It proposes potential issues in the rehabilitation processes of contaminated ecosystems, offering valuable insights for the formulation of environmental pollution control strategies. The paper is generally well-written; however, there are a few minor issues that need attention.

Response

Thank you so much for your opinions and comments. We have done our best to take them into account to improve the manuscript as much as possible.

I recommend considering acceptance after addressing the following:

  1. Modify some minor language errors.

Response

Thank you, the edits have been made.

  1. Ensure uniformity in the reference format, particularly in lines 166, 196, 260, and so on.

Response

The format of the citation was done in a uniformed manner. Thank you for pointing this out.

Dear Reviewer,

Thank you again for reviewing our manuscript, and we appreciate your feedback and time dedicated to improving our manuscript.

Reviewer 3 Report

Comments and Suggestions for Authors

This paper Identified the advantages and disadvantages of using biosorption technologies for the joint fixation of heavy metals and radionuclides, and analyzed the sustained possibility of using phosphogypsum for soil bioremediation. Overall, this manuscript was written well and of novelty. My main comments are as follow:

1. Heavy metal contaminated soil can be safely utilized by applying amendments (e.g. Environ Sci Pollut Res 2020, 27: 27859–27869; Environ Sci Pollut Res 2020, 27: 39391–39401). Therefore, the authors should refer to more literature.

2. Providing soil with nutrients it lacks is a promising method that helps plants resist heavy metal stress (e.g. Sci Total Environ 2023, 903:166264; Ecotoxicol Environ Safe, 2023, 250: 114501).

Author Response

Responses to reviewer comments 3

This paper Identified the advantages and disadvantages of using biosorption technologies for the joint fixation of heavy metals and radionuclides, and analyzed the sustained possibility of using phosphogypsum for soil bioremediation. Overall, this manuscript was written well and of novelty.

My main comments are as follow:

  1. Heavy metal contaminated soil can be safely utilized by applying amendments (e.g. Environ Sci Pollut Res 2020, 27: 27859–27869; Environ Sci Pollut Res 2020, 27: 39391–39401). Therefore, the authors should refer to more literature.
  2. Providing soil with nutrients it lacks is a promising method that helps plants resist heavy metal stress (e.g. Sci Total Environ 2023, 903:166264; Ecotoxicol Environ Safe, 2023, 250: 114501).

Response

Thank you for your comments, we have made appropriate additions to the text of the manuscript:

Summarizing a number of studies also two directions of soil management can be noted such as:

  1. Soil additives can be used to fix toxicants (e.g. Environ Sci Pollut Res 2020, 27: 27859-27869; Environ Sci Pollut Res 2020, 27: 39391-39401).
  2. Providing the soil with nutrients it is deficient in helps plants resist stress caused by heavy metals (e.g., Environ Total Environ 2023, 903:166264; Ecotoxicol Environ Safe, 2023, 250: 114501).

Dear Reviewer,

Thank you again for reviewing our manuscript and we appreciate your feedback and time dedicated to make our manuscript better.

Reviewer 4 Report

Comments and Suggestions for Authors

After carefully reading your article, we have noticed the following issues:

1. Insufficient results and discussion section: The presentation of the results and discussion in your article appears to be inadequate. You failed to clearly explain the relationship between radioactive pollution and heavy metal pollution, as well as effectively demonstrate the feasibility of using phosphogypsum for soil bioremediation. We suggest that you better develop each section in conjunction with the objectives stated in the introduction, clearly state the problems to be addressed in each section, highlight the viewpoints to be expressed in each section, and increase the specificity and depth of the results and discussion sections.

2. Literature listing issue: We found extensive citations of literature in your article, which did not effectively address the issues to be discussed. We recommend that you extract your own viewpoints from these references and better illustrate the points to be made in the article, enhancing its practicality and readability.

3. Sentence logic unclear: We have noticed that many sentences in your article contain logical issues and improper use of transitional words. We suggest that you reorganize and adjust the sentence structure to make the sentence logic and article logic clearer and more coherent, while using concise and accurate language to express your ideas.

We regret to inform you that we are unable to accept your manuscript.

Comments on the Quality of English Language

There are many issues with the improper use of conjunctions in sentences in the article, resulting in unclear logical relationships between sentences. It is recommended that you reorganize and adjust the sentence structure, and try to use concise and accurate language to make the logic of the sentences clearer and more coherent.

Author Response

Responses to reviewer comments 4

After carefully reading your article, we have noticed the following issues:

  1. Insufficient results and discussion section: The presentation of the results and discussion in your article appears to be inadequate. You failed to clearly explain the relationship between radioactive pollution and heavy metal pollution, as well as effectively demonstrate the feasibility of using phosphogypsum for soil bioremediation. We suggest that you better develop each section in conjunction with the objectives stated in the introduction, clearly state the problems to be addressed in each section, highlight the viewpoints to be expressed in each section, and increase the specificity and depth of the results and discussion sections.

Response

Thank you very much for your critical view, we have deepened the analysis of previous studies and expanded the number of references. Changes have been made in the Results and Discussion section for a better understanding of the information. The article discusses the important issue of remediation of soils contaminated with heavy metals and radionuclides, since these toxicants are present simultaneously in contaminated areas. Therefore, remediation techniques should take into account the specificity of both of them.

  1. Literature listing issue: We found extensive citations of literature in your article, which did not effectively address the issues to be discussed. We recommend that you extract your own viewpoints from these references and better illustrate the points to be made in the article, enhancing its practicality and readability.

Response

We have reviewed the literature in detail and also provided additional references to delve deeper into the topic of the review.

  1. Sentence logic unclear: We have noticed that many sentences in your article contain logical issues and improper use of transitional words. We suggest that you reorganize and adjust the sentence structure to make the sentence logic and article logic clearer and more coherent, while using concise and accurate language to express your ideas.

Response

The structure of the proposals has been reorganized and clarified. All changes to the text are highlighted in green.

We regret to inform you that we are unable to accept your manuscript.

Comments on the Quality of English Language

There are many issues with the improper use of conjunctions in sentences in the article, resulting in unclear logical relationships between sentences. It is recommended that you reorganize and adjust the sentence structure, and try to use concise and accurate language to make the logic of the sentences clearer and more coherent.

Response

Thank you so much for your opinions and comments. We have done our best to take them into account to improve the manuscript as much as possible. We had our manuscript also reviewed by a native speaker (using British English).

Dear Reviewer,

Thank you again for reviewing our manuscript, and we appreciate your feedback and time dedicated to improving our manuscript.

Reviewer 5 Report

Comments and Suggestions for Authors

This work is a review covering heavy metal and radionuclides contamination, related bioremediation approaches. There are a lot of issues for the authors to address. Here they are:

1) the designed structure of the paper is quite confusing or unique. It is a review but the authors also added their own data on phosphogypsum components, making it a hybrid of review paper and a research paper. If it is a review, the authors obviously did not take it seriously as Table 2, Table 3, figure 7 and other figures mainly include 2-5 cited papers and the summary contains no data at all just some abstract description. It is not objective at all and no value for other researchers to refer to. If it is a research paper, only very limited data is shared in Figure 8, far less than required for the type in this journal.

2) Title needs to be revised. The whole manuscript, including the abstract is talking about heavy metal and radionuclides contamination and related remediation. It is a very specific, narrowed field while your title is very broad. Please revise your title and make it cover the specific topic. 

3) Phosphogypsum is well known for its radioactivity due to the presence of radon emission and related risk to cause cancer by US EPA, it is not suggested to add one radioactive material to the radionuclides polluted soil for the remediation purpose unless you can provide reliable data to support it. The design of this manuscript is not so reasonable in my opinion. 

4) conflicting or confusing description across the manuscript. For example, In Figure 4 and Table 3, the definition and the applications of "Biosorption" is so confusing and inconsistent. In the definition you explained it as using cells to remove metals while in the process features you mentioned biomass, sorbent materials. It is so conflicting. 

5) In the Figure 6, the caption is "Influence of the organic-mineral complex on the fixation of heavy metals and radionuclides in soils." while the photo used to support your caption is not related at all. There is no logic between these two items. 

Comments on the Quality of English Language

The whole manuscript is full of grammar error, improper, confusing and conflicting description. Logic is poor in the manuscript and make it so hard to read. Here are some examples: Line 32, page 1, it is not so objective to claim radioactive & heavy metal contamination the most dangerous types of pollution. Please revise it. Line 39, page 1, negative 5%? or around 5%? Very confusing and please revise it with a "~". Line 47, page 2, Grammar error, please delete it as there is no cause-and-result logic here. Line 50, page 2, Please check the description here as it is very confusing. Do you mean 134-137Cs? Line 51, Please specify the size of the particles for readers to have a better and easier understanding. Line 74 very confusing and not friendly for others to read. Line 77-78 when claim something or discovery with the "highest" or "most" or "best", you have to be very careful. To make your argument count or convincing, better to remove these words or you need to add 3-5 more citations to support it. Only [7] is not enough. This is science with the highest level of objectiveness required, not novels. I noticed similar issues kept occurring in this manuscript. Please be objective. 

There are many similar description or grammar errors across the manuscript, please revise it.

Author Response

Responses to reviewer comments 5

This work is a review covering heavy metal and radionuclides contamination, related bioremediation approaches. There are a lot of issues for the authors to address. Here they are:

  • the designed structure of the paper is quite confusing or unique. It is a review but the authors also added their own data on phosphogypsum components, making it a hybrid of review paper and a research paper. If it is a review, the authors obviously did not take it seriously as Table 2, Table 3, figure 7 and other figures mainly include 2-5 cited papers and the summary contains no data at all just some abstract description. It is not objective at all and no value for other researchers to refer to. If it is a research paper, only very limited data is shared in Figure 8, far less than required for the type in this journal.

Response

Thank you very much for your comments. Table 2 and 3 have been updated with references. Figures 2-7 are author's figures, but we have expanded the references to previous studies in the text.

We have added data from our own studies that are consistent with previous studies in this area of phosphogypsum application, which we think is appropriate. But thank you very much for your critical view, deepened the analysis of previous studies and expanded the number of references.

Regarding the structuring of our manuscript. We have made a change in the Results and Discussion section for better perception of the information:

3.1. Review of the state of ecosystems contaminated with heavy metals and radionuclides

3.1.1 Sources of radionuclides and heavy metals in the ecosystem

3.1.2 Monitoring of radionuclides and heavy metals in ecosystems and impact on humans: Ukraine case study

3.2 Biotechnologies for integrated fixation of heavy metals and radionuclides: identification of advantages and disadvantages

3.2.1 Soil bioremediation methods

3.2.2 Biosorption technologies and their realization aspects

2) Title needs to be revised. The whole manuscript, including the abstract is talking about heavy metal and radionuclides contamination and related remediation. It is a very specific, narrowed field while your title is very broad. Please revise your title and make it cover the specific topic. 

Response

Thank you very much for your comment we have made a change to the title of the article: Soil contamination by heavy metals and radionuclides and related bioremediation techniques: A review

3) Phosphogypsum is well known for its radioactivity due to the presence of radon emission and related risk to cause cancer by US EPA, it is not suggested to add one radioactive material to the radionuclides polluted soil for the remediation purpose unless you can provide reliable data to support it. The design of this manuscript is not so reasonable in my opinion. 

Response

Thank you very much for your comment, we have made a clarification on the manuscript text:

... it is noteworthy that phosphogypsum may be contaminated with radionuclides [129]. According to EPA data, phosphogypsum contains significant quantities of uranium and its decay products, such as radium-226, attributed to their presence in phosphate ores. The concentration of uranium in phosphate ores identified in the United States varies within the range of 0.26 to 3.7 Bq/g (7 to 100 pCi/g) [130]. However, diverse raw materials are utilized in different countries and regions globally; consequently, not all phosphogypsum exhibits elevated radioactivity levels [131, 132].

  • conflicting or confusing description across the manuscript. For example, In Figure 4 and Table 3, the definition and the applications of "Biosorption" is so confusing and inconsistent. In the definition you explained it as using cells to remove metals while in the process features you mentioned biomass, sorbent materials. It is so conflicting. 

Response

We have concretized the definition of biosorption

Biosorption is a physicochemical and metabolically independent process based on various mechanisms including uptake, adsorption, ion exchange, surface complexation and precipitation.

Ref

Marina Fomina, Geoffrey Michael Gadd, Biosorption: current perspectives on concept, definition and application, Bioresource Technology, Volume 160, 2014, Pages 3-14, https://doi.org/10.1016/j.biortech.2013.12.102.

It is worth noting that this is a rather extensive term that has a rich interpretation, so we tried to take into account the most important features of it.

  • In the Figure 6, the caption is "Influence of the organic-mineral complex on the fixation of heavy metals and radionuclides in soils." while the photo used to support your caption is not related at all. There is no logic between these two items. 

Response

 Thank you for your comment, the title does not refer to the figure itself, it was a technical error, we have corrected it.

Fig. Accumulation of phosphogypsum in the environment

It should be noted that the numbering of the figures has been changed.

Comments on the Quality of English Language

The whole manuscript is full of grammar error, improper, confusing and conflicting description. Logic is poor in the manuscript and make it so hard to read. Here are some examples:

Line 32, page 1, it is not so objective to claim radioactive & heavy metal contamination the most dangerous types of pollution.

Thanks for the comment, a clarification has been made:

Therefore, the one of most dangerous types of pollution associated with radioactive contamination ....

In addition, the whole manuscript has been reviewed by a native English speaker to polish the text.

Please revise it. Line 39, page 1, negative 5%? or around 5%? Very confusing and please revise it with a "~".

Response

Thank you, this is a technical error, has been corrected

Line 47, page 2, Grammar error, please delete it as there is no cause-and-result logic here.

Response

Thank you, we've made the correction:

In [4] was presented research in areas affected by nuclear power plant (NPP) disasters. Thus, 31 radioactive particles from surface soils were detected in an area 3.9 km northwest of the Fukushima-1 NPP.

Line 50, page 2, Please check the description here as it is very confusing. Do you mean 134-137Cs?

Response

134 + 137Cs had the highest activity ever recorded for Fukushima-1 NPP (6.1×105 and 2.5×106 Bq per particle after decay correction until March 2011).

Line 51, Please specify the size of the particles for readers to have a better and easier understanding.

Response

Thank you for your comment, the size has been clarified:

Considering their large size (120 μm)...

Line 74 very confusing and not friendly for others to read.

Thank you, we've made the correction.

Line 77-78 when claim something or discovery with the "highest" or "most" or "best", you have to be very careful. To make your argument count or convincing, better to remove these words or you need to add 3-5 more citations to support it. Only [7] is not enough. This is science with the highest level of objectiveness required, not novels. I noticed similar issues kept occurring in this manuscript. Please be objective. 

Response

Thank you, we've made the correction:

According to the sorption efficiency of these isotopes, the soil is arranged in the following order: sod-podzolic soils (Albeluvisols), gray soils (Calcisols), then yellow soils, red soils (Ferralsols, Alisols, Acrisols), chestnut soils (Kastanozems) and black soils (Chernozem). A substantial transfer of radiocaesium to plants in sandy and sandy loam soils with a low content of clay minerals and organic matter has been reported [7].

There are many similar description or grammar errors across the manuscript, please revise it.

Response

Thank you so much for your opinions and comments. We have done our best to take them into account to improve the manuscript as much as possible.

Dear Reviewer,

Thank you again for reviewing our manuscript, and we appreciate your feedback and time dedicated to making our manuscript better.

Round 2

Reviewer 5 Report

Comments and Suggestions for Authors

The work has been improved based on my comments. However, as a review for soil remediation (heavy metal and radionuclides), the quality is low and have not so much contribution to the field. For example, the whole manuscript is about related bioremediation techniques for heavy metal and radionuclides removal from soil. For the techniques, the key part is the specific data from the previous published papers or cases to show how effective each technique or method is (e.g., after 1 year treatment, 50% heavy metal are removed or 50% less heavy metal can be rinsed out from soil suggesting the method can stabilize the heavy metals from entering plants, surface water or groundwater).  The authors did some revision on the tables and figures, but mainly by simply adding more references without considering the specific data I suggest them to add. Thus, this article does not match the standards required by the journal in my opinion.